# Machine Learning-Based Structural Health Monitoring Using RFID for Harsh Environmental Conditions

**Aobo Zhao** [1,2] , **Ali Imam Sunny** [1] , **Li Li** [1,*] **and Tengjiao Wang** [2]

1   Nuclear Advanced Manufacturing Research Centre, iHub, Infinity Park Way, Derbyshire DE24 9FU, UK; aobzhao@seeneuro.com (A.Z.); ali_sn@hotmail.com (A.I.S.)
2   Seeneuro Ltd., 4th Floor, Building 4, No. 85 Keji Avenue, Yuhang Street, Hangzhou 310000, China; wangtj@seeneuro.com
*   Correspondence: li.li@namrc.co.uk

**Abstract:** Post Operation Clean Out (POCO) is the process to remove hazardous materials and decommission nuclear facilities at the end of a nuclear plant's lifetime. The introduction of Internet of Things (IoT) technologies in the environment, especially radio frequency identification (RFID), would improve efficiency and safety by intelligently monitoring POCO activities. In this paper, we present a passive material identification and crack sensing method developed for the integration of sensing and communication using commercial off-the-shelf (COTS) RFID tags, which is a long-term solution to material property monitoring under insulation for harsh environmental conditions. To validate the effectiveness of material identification and crack monitoring, machine learning techniques have been applied, and the feasibility of the study has been outlined. The result shows that the material identification can be achieved with traditional features and obtain improved accuracy with three-layer multi-layer neural networks (MLNN). In crack characterization, the tree algorithm based on traditional features achieves a reasonable accuracy, while three-layer MLNN is the best solution, which supports the efficiency of traditional feature extraction methods in specific applications.

**Keywords:** structural health monitoring; RFID; machine learning; non-destruction testing; nuclear decommissioning

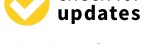



## 1. Introduction

Non-destructive testing (NDT) methods are used in industry to evaluate the integrity and properties of material or components without destroying the tested object. The IAEA promotes the use of non-destructive testing technology to maintain the stringent quality control standards for the safe operation of nuclear and other industrial installations [1]. One of the most common types of defects are cracks, whose detection in time are a prerequisite to smooth operation and the prevention of future failures [2]. With the development of sensors and the Internet of Things (IoT), defect detection and characterization techniques are enabled by embedding sensors for large-scale infrastructure. The ability to inspect defects or classify them without the shutdown of industry for unscheduled time is an important economic consideration [3–5], and therefore, a monitoring method with predictive maintenance capability is highly desirable.

Many NDT&E methods can be used for high-temperature inspection and monitoring, such as laser ultrasonic [6,7], thermal infrared imaging technology [8] and laser electromagnetic acoustic transducer (EMAT) configurations [9–11], the MFL (magnetic flux leakage) method for crack monitoring [12], and even eddy current and pulsed eddy current techniques [13,14]. However, these inspection techniques come with complexity and costs for long-term measurements in condition-monitoring applications. Additionally, these techniques have their weaknesses, particularly in remote locations with limited access, since their usage is dependent on cabling or battery-operated electronics. Thus, new technologies for the deformation monitoring of conveyor belt structures are required. Most recently,

radio frequency identification (RFID)-based sensors have been proposed as a device for remotely gathering information about crack detection and its growth [15–18].

The benefit of RFID is that it can operate in high temperature conditions [19] for its passive nature. The chip-based RFID can operate at 160 degrees [3] without the constraint of batteries, while the chip-less RFID is able to tolerant hundreds of degrees without an integrated circuit (IC). However, in chip-less tags, the anti-collision techniques are totally dependent upon the reader, whereas in chip-based tags, the IC is dedicated for tagging purposes. The chipped passive wireless RFID sensors have gained much attention in both academies and industries for potential SHM [2,18]. Low-frequency (LF) and high-frequency (HF) RFID sensors have been used previously to characterize the steel corrosion progression [17,20,21], and the read range was later improved by the usage of 3D ultra-high-frequency (UHF) band RFID [22,23]. However, communication and sensing in LF/HF RFID systems are antithetical, where the former is based on magnetic resonance coupling (MRC) and the latter relies on magnetic inductive coupling. Though the time-domain-based RFID system has an immense communication speed over frequency domain [24], the latter is advantageous in terms of working at different resonant frequencies.

UHF RFID systems can be easily installed, have a large reading range, and do not require dedicated line-of-sight access [25]. Moreover, in RFID technology, each sensor has its identity, so an array configuration of multiple sensors can be distributed over the structure to be monitored, and each sensor in this array can be easily identified with its location on the structure/material. This type of focused monitoring system increases the ability to keep track of potential hot-spots and provides early opportunity to inspect and fix possible damages from worsening [26]. By using a passive RFID sensor, it is possible to obtain the status data (e.g., temperature, stress, crack, etc.) of a structure without disassembling it [3,27]. However, one of the major challenges arises when there are multiple parameters to be inspected in harsh environments such as nuclear power plants or decommissioning sites. It is more evident that it is vital and very much desirable to have an automated system or application. Therefore, incorporating machine learning (ML)-based health monitoring will vastly improve the system's performance. Installing an algorithm based on ML will eliminate the uncertainties involved with human supervision. It can work with multi-dimensional data and is able to improve the model by self-learning [28,29]. The last two decades have seen a rise in the use of ML methods for handling several identification problems in engineering domains [30]. However, structural health monitoring within the nuclear industry using ML-incorporated UHF RFID sensors is still not addressed comprehensively in the literature. Machine learning-based crack detection will help allow the system to take place in real time, significantly improving the system's performances [31].

UHF RFID sensors have been used for crack detection for many applications. An RFID-based sensor is used for concrete crack detection in [32]. Here, the sensor operates in conjunction with another attached conductive surface, and the strain of the concrete causes the electrical resistance to be increased. Therefore, the authors are able to identify damage from the tag response. In [33], RFID-based sensors are proposed to depict gradual metallic plates' crack damage. The results in [32] show that the RFID sensors are able to detect crack widths of as low as 0.0650 mm. An investigation was conducted in [34,35] for the crack depth over stainless steel and ferromagnetic materials by combining RFID sensors with the ThingMagic platform. A UHF RFID tag is proposed in [33] to monitor the tyre's health, where a modified end loaded meander line dipole antenna is used to compensate for the decreased read range. However, the existing studies of RFID-based sensors in the literature such as [32–35] do not consider the significant prediction outcomes when combined with the ML algorithms.

The implementation of ML techniques is vital for dynamic operations of the systems with continuous and automated learning [36]. ML algorithms are generally divided into supervised and unsupervised types, which are analysed for the learning and prediction of empiric results. Supervised learning algorithms minimise the error between the targeted data and output data, whereas unsupervised algorithms are adopted for clustering data

when data training is not preferable. However, both types of ML algorithms can be utilised for the material classifications based on the site conditions and circumstances for the availability of input data [37]. Construction material classification via ML techniques has gained a lot of attention among professionals and researchers in the construction sector. Various studies can be found related to material classification for construction progress monitoring. However, still, improvements are required in the methodologies and algorithms towards effective and efficient outcomes. Other than pattern recognition, ML technologies are adopted for the self-learning of the big-data based systems connected via the Internet of Things (IoT) integrated with digital technologies [38]. Likewise, for the construction progress detection technologies, the trend of integration with ML techniques for the digitalisation of the monitoring process has also been increased in recent times [39]. ML has different recognized branches such as artificial neural networks (ANN), support vector machines (SVM), and fuzzy logic-based systems (FS) [30]. ML-based crack detection methods are widely used for civil applications to detect cracks in pavements using deep neural networks [40], concrete surfaces incorporating fuzzy logic and ANN [41], and steel bridge girders using SVM in [30].

The focus of the paper is to explore the capabilities of ML in the nuclear sector, and this is performed by carrying out a comparative study of identifying the correct material based on the ML approach and then determining varying cracks based on width, depth, and length using both a multilayer neural network (MLNN) and classic feature-based classification. The concept of this model involves an MLNN and its input features in terms of the backscattered power, transmitted power, electric field strength, and unwrapped phase, which are obtained from an RFID sensor. In order to identify the significance and future impact of the crack, the paper furthermore expands its objective by performing the detection of both materials and cracks of different sizes. Such modelling essentially leads to a highly accurate system capable of identifying cracks and crack widths. Consequently, the UHF RFID sensor response can be proficiently interpreted to monitor infrastructure health and the decommission of the old reactors, which is unavoidable at the end of their operation, and POCO is carried out to tackle the radiation contaminated materials and ensure the safety and efficiency in the process.

## 2. Proposed Methodology

### 2.1. RFID Base Material Identification and Crack Sensing

The principle of RFID-based sensing can be expressed as follows. RFID is based on backscattering theory, which enables passive information exchange, as shown in Figure 1a. The communication is established with a switchable load inside the RFID chip. When the RFID tag antenna harvests the electromagnetic wave from the reader and powers up the chip, the chip switches the load to change the chip impedance, which causes a change in the matching condition between the tag antenna and tag chip, leading to an amplitude shift keying (ASK) modulated backscattered signal [42]. When the tag is placed on a specific material, as shown in Figure 1b, the dielectric constant of the material changes the impedance of the tag antenna, which further changes the matching status, resulting the change in various communication features, such as transmitted power, backscattered power, phase, etc. These features thus contain information about the dielectric constant of the material, which provide potential for material identification with RFID technology.

For simple antennae such as microstrip line antennae or meander line antennae, which are normally used in RFID tags, the effect of placing a dielectric material can be modelled as inserting a layer with a dielectric constant of $\varepsilon_{eff}$ and loss tangent $\delta_{eff}$ under the tag, which can be regarded as a parasitic capacitor and a parasitic resistor to the tag antenna [43]. This parasitic capacitor adds a negative reactance to the tag antenna impedance, and the parasitic resistor increases the antenna resistance, thus causing a mismatch between the tag antenna and the tag chip, reflected in a decrease in transmitted power and back-scattered power.

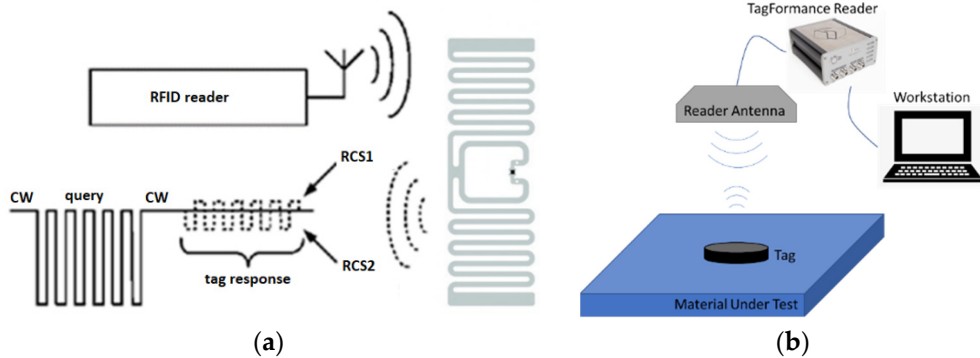

**Figure 1.** (**a**) RFID principle, (**b**) RFID-based material identification and crack sensing system.

The measurement for effective permittivity is not straight forward but can be analysed with the model presented in [44]. Assuming that the transmission parameter from the reader antenna to the tag antenna is $S_{12}$ and back-scattering transmission parameter is $S_{21}$ ($S_{12}$ is the reverse voltage gain whereas $S_{21}$ is the forward voltage gain), the transmission parameter from the tag antenna to the tag chip is $S_{23}$, and the transmission from tag chip to antenna is $S_{32}$, the equation describing the transmission is:

$$S_{13} = S_{12}S_{23} \tag{1}$$

After inserting the dielectric layer, the transmission parameter is:

$$S'_{13} = S_{12}S'_{23} = S_{12}S_{23} \times \frac{Z_{ic}}{Z_{ant} + Z_{ic} - \frac{1}{j\omega C_p}} \times (1 - tan\delta) \tag{2}$$

$$= S_{13} \times \frac{Z_{ic}}{Z_{ant} + Z_{ic} - \frac{1}{j\omega C_p}} \times (1 - tan\delta)$$

The parasitic capacitance is defined by $\varepsilon_{eff}$ as

$$C_p = \frac{\varepsilon_{eff}S_{eff}}{d_{eff}} \tag{3}$$

where $S_{eff}$ and $d_{eff}$ are effective area and distance for an ideal planar capacitor model [45].

In our study, the experiment setup and sample diameter are the same; thus, $S_{13}$, $S_{eff}$, and $d_{eff}$ are the same for different dielectric materials. In this way, the change of transmitted power is a direct measure of the material dielectric property $\varepsilon_{eff}$ and the loss tangent $\delta$. Similarly, backscattered power and phase are all directly related with $\varepsilon_{eff}$ and $\delta$, providing the capability of identifying the dielectric material.

A metal model is more complicated as it introduces parasitic inductance apart from resistance and capacitance. The transmission equation is:

$$S''_{13} = S_{13} \times \frac{Z_{ic}}{Z_{ant} + Z_{ic} - \frac{1}{j\omega C_p} + jwL_p} \times (1 - tan\delta) \tag{4}$$

Thus, the measurement for the crack requires more parameters for measurement.

### 2.2. Machine Learning-Aided Material Identification

Original feature analysis methods are based on raw features such as received signal strength intensity [46] and phase [47]. Benefits of these features are that they are easily accessible, and initial research shows the effectiveness of their features. However, these features suffer from environmental interference and are generally relative measurements rather than absolute values of the material property. Further investigation has focused on

various feature extraction methods, such as principal component analysis, tree, or Naïve Bayes. As the operation environment of RFID is rough, and backscattered signal is weak and vulnerable, these feature extraction methods suffer from low accuracy.

ML has a shown superior advantage in analysing multi-modal and multi-dimensional datasets. To analysis and validate the effectiveness of ML in RFID-based sensing, we propose a multi-layer neural network-based classification method that uses the supervised training with an algorithm known as error back propagation to perform tasks. The structure of an MLNN can be shown as Figure 2. The MLP develops a mapping function between the inputs and outputs, including several hidden layers in between. The learning process is that the processing elements (PEs) in the input layer receive data, such as transmitted power, backscattered power, etc., then pass them to the PEs in the hidden layer. A simple mathematical computation for extracting the weight of the links is undertaken by the PEs. The results from the hidden PEs are mapped onto appropriate threshold function of each PE, and the outputs are produced as the input to the next layer. Upon reaching the output layer, the final results are compared with actual parameters to evaluate the effectiveness, and the processing cost is calculated for measuring the efficiency.

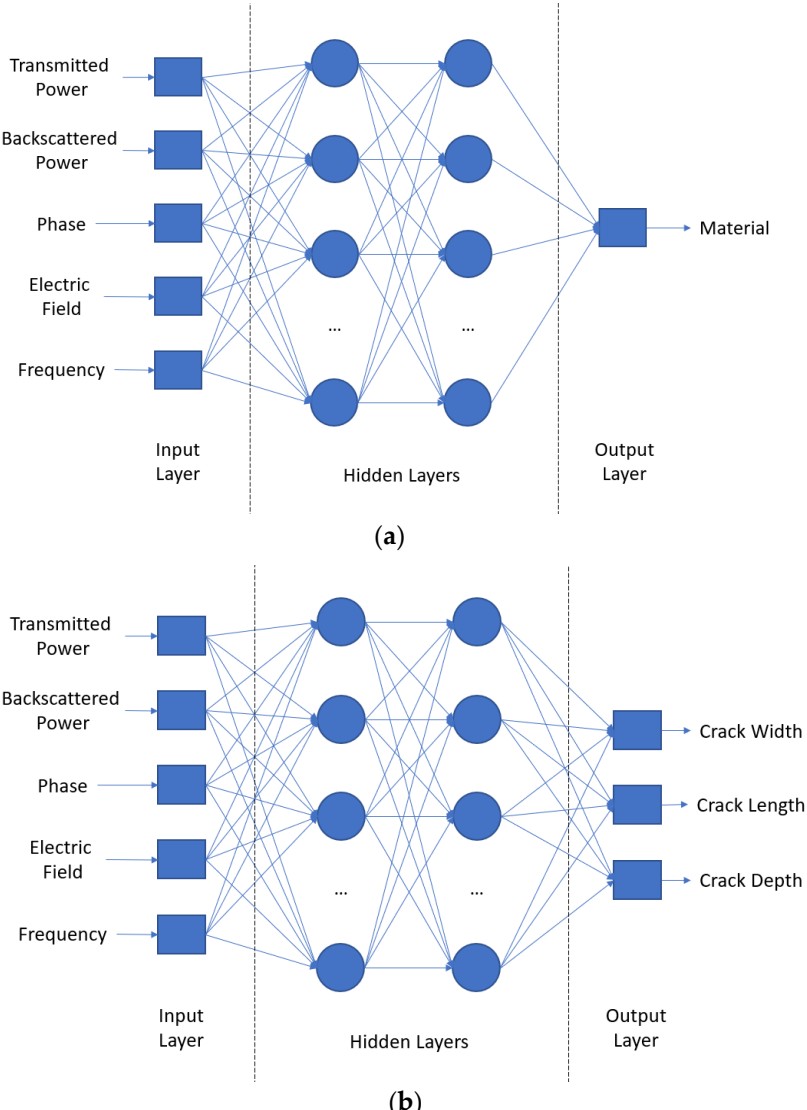

**Figure 2.** MLNN structure for RFID-based sensing: (**a**) material identification MLNN and (**b**) crack sensing MLNN.

### 2.3. Material Sample Preparation

The materials used in this experiment are shown in Table 1. Figure 3 depicts the real materials. The selected materials include groups of low-permittivity, low-loss material (PTFE); low-permittivity, low-medium material (PVC); low-permittivity, high-loss material (POM); low-permittivity, extremely high-loss material (cardboard); medium-permittivity, medium-loss material (FR-4); and high-permittivity, medium-loss material (rubber). Thus, we can have a group of low-permittivity materials to compare the effect of material loss to the efficiency of wireless power transmission and a group of medium-loss materials to compare the effect of permittivity.

**Table 1.** Material list.

| Material | Relative Permittivity | Dielectric Loss Tangent |
|---|---|---|
| Cardboard | 2.57 | 0.0717 |
| FR-4 | 4.87 | 0.0141 |
| Glass | 7.11 | 0.0098 |
| Polyoxymethylene (POM) | 2.96 | 0.0450 |
| Polytetrafluoroethylene (PTFE) | 2.05 | 0.0002 |
| Polyvinylchloride (PVC) | 3.00 | 0.0079 |
| Rubber | 6.73 | 0.0247 |

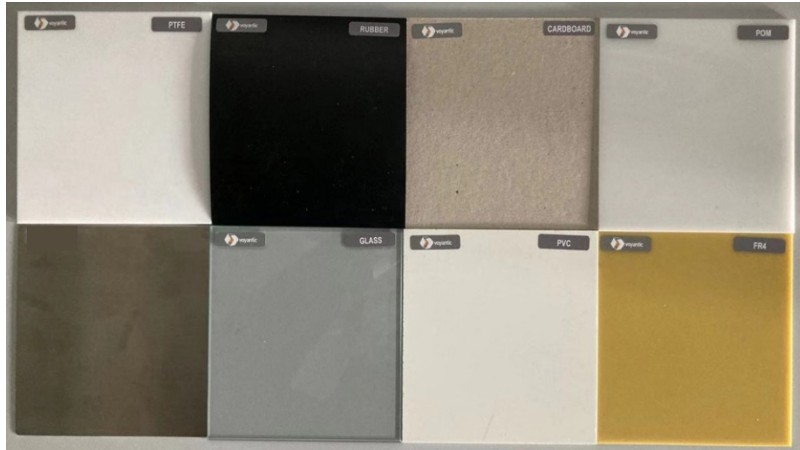

**Figure 3.** Photo of the materials used for experimental studies.

### 2.4. Crack Information

Stainless steel and carbon steel are essential materials in the nuclear industry for their superior strength characteristics and cost efficiency. However, all metals face the challenge of cracks either from long-term fatigue or extreme force applied. RFID-based crack sensing could reduce the human power for undertaking routine examination and improve effectiveness in the long-term monitoring of defect growth. However, the challenges are the instability of RFID signal features and environmental noise. Here, we evaluated artificial cracks on two different metal materials for their capability in crack sensing and the effectiveness of ML compared with classic features. Two samples, carbon steel and a stainless steel, had the same size of $300 \times 300 \times 20$ mm$^3$. Artificial cracks were created by drilling slots on the samples. The specifications of the slot cracks are given in Figure 4c, where the unit is millimetre. They consisted of three series of cracks, which represent the increase of depth, width, and length, individually. Figure 4a,b are the images of the samples with cracks.

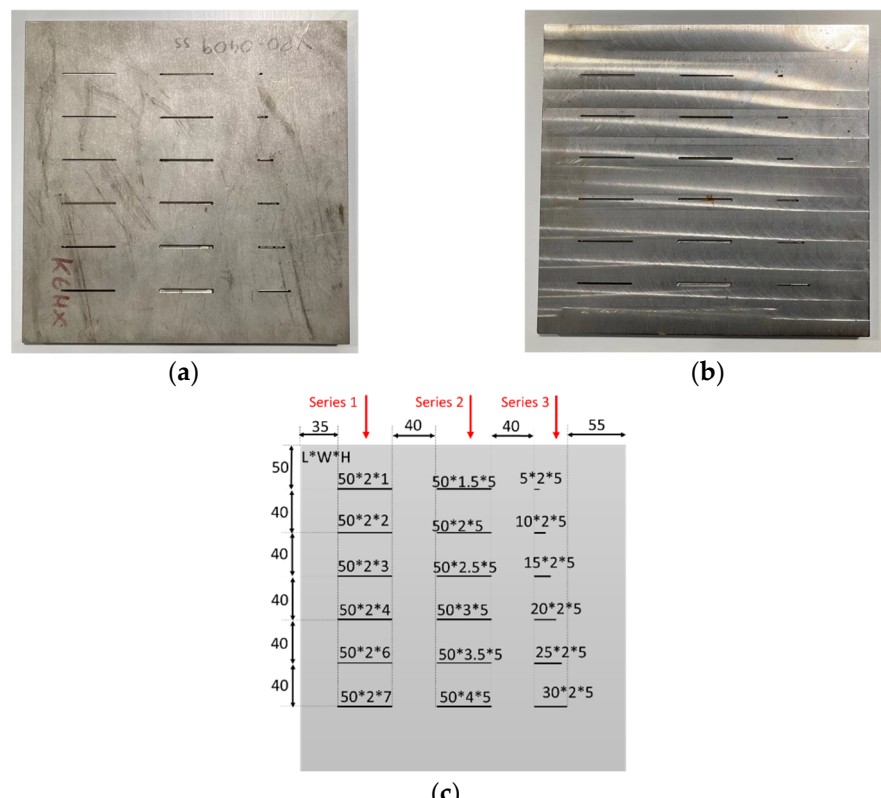

**Figure 4.** (**a**) Stainless steel; (**b**) carbon steel; and (**c**) crack sample preparation schematic.

### 2.5. Data Preparation and Accuracy Equation

The results were pre-processed for ML as the multi-model and multi-dimensional dataset was used. For each crack, it contained measurements of the transmitted power (TP), backscattered power (BP), phase (PH), and electric field (EF) for the frequency range between 800 MHz and 1000 MHz. This dataset was reorganised into an N-by-M matrix so that the MLNN could evaluate where N was the observation number and M was the feature number. Each row represents a single measurement that contains the measurement frequency, measured TP, BP, PH, and EF, as well as the crack dimension and the sample material. Seventy percent of the overall dataset was used for training the MLNN, and the remaining thirty percent was used for validating the results.

After inputting the data into the MLNN, the number of neurons, number of iterations, and activation functions were investigated to obtain an optimal model to extract the target feature, which is the material for the material identification and crack parameters for crack sensing. The accuracy equations for assessing the performance of the model for material identification and crack sensing were defined as

$$Accuracy = \frac{\sum Correctly\ Identified\ Material\ Data}{Total\ dataset\ number}(Material\ Identification) \tag{5}$$

$$Accuracy = \frac{\sum Correctly\ Measured\ Crack\ Data}{Total\ dataset\ number}(Crack\ Sensing) \tag{6}$$

## 3. Experiment Validation

### 3.1. Experiment Setup

The validation experiment was carried out by measuring the tag signal and extracts of the features when the tag was placed on different materials or defects, as shown in Figure 5. The system consisted of a transponder as the sensor, a reader with an antenna as a transceiver to power the tag and acquire the signal, and a laptop to control and collect data from the reader. A UHF RFID development kit from Voyantic Tagformance Pro was

used as the reader in the validation experiment. The reader antenna was connected to the development kit using a directional coupler, which enables the separation of tag signals and reader signals. The antenna was held using a tripod with a distance of 40 cm, so that the tag response was measurable over the entire band from 800 to 1000 MHz. The tag was placed on the target material sample with the polarization direction aligned with the reader antenna. There were two green markers on the bench to ensure that the position of the sample was the same for different materials.

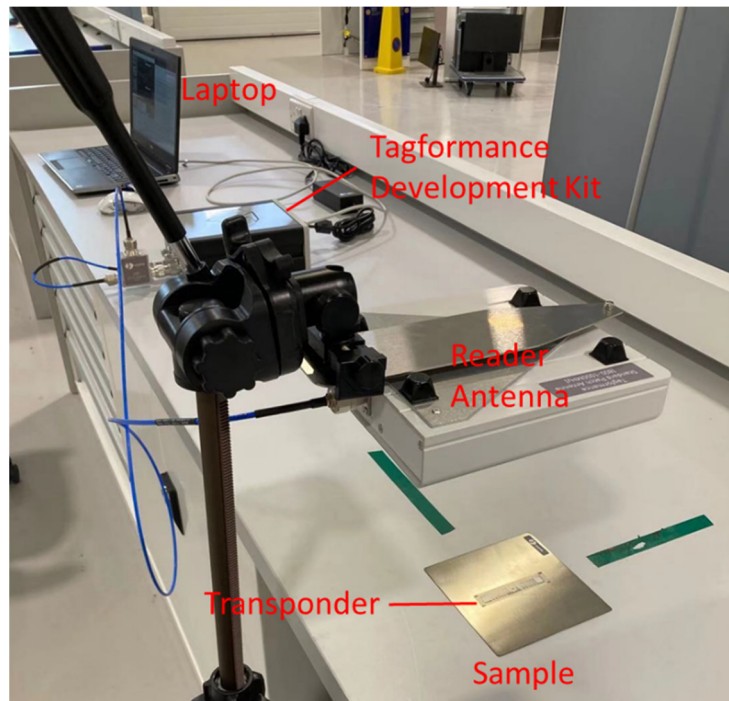

**Figure 5.** Experiment setup for RFID-based sensing system.

### 3.2. COTS RFID Tags for Validation

The selected UHF transponders are shown in Figure 6. Transponders a, b, and c were anti-metal tags. a was a dipole-based tag, b was a patch antenna tag, and c the coil antenna-based tag. Transponder d as a commonly used commercial UHF tag using a dipole antenna design. e as the reference tag from Voyantic used to calibrate the Tagformance UHF RFID development kit. The specifications are listed in Table 2.

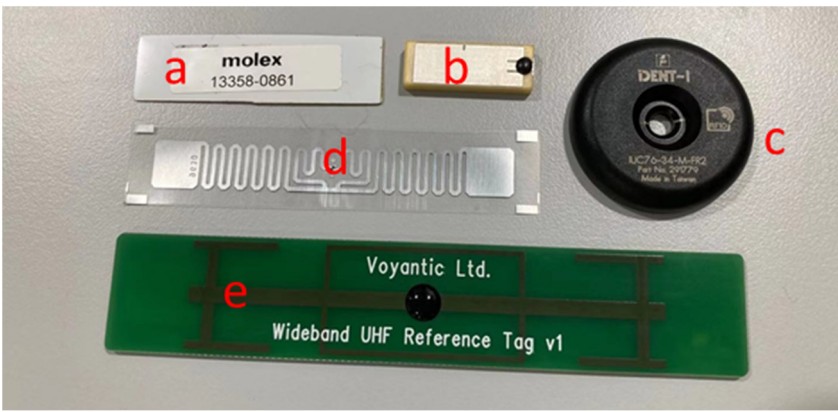

**Figure 6.** Experiment setup for RFID-based sensing system.

**Table 2.** Specifications of selected transponders.

| Transponder | Size (mm) | Read Range (m) | Operation Band (MHz) |
|:---:|:---:|:---:|:---:|
| a | $49.8 \times 14 \times 0.4$ | 3–3.6 | 865–928 |
| b | $25 \times 9 \times 3.7$ | 1.8 | 902–928 |
| c | Round $34 \times 34 \times 6$ | NA | 902–928 |
| d | $80 \times 16 \times 0.1$ | NA | NA |
| e | $116 \times 22 \times 0.9$ | NA | 800–1000 |

### 3.3. Validation Results

The first step was the validation of the RFID-based crack sensing using traditional features. The results are shown in Figure 7.

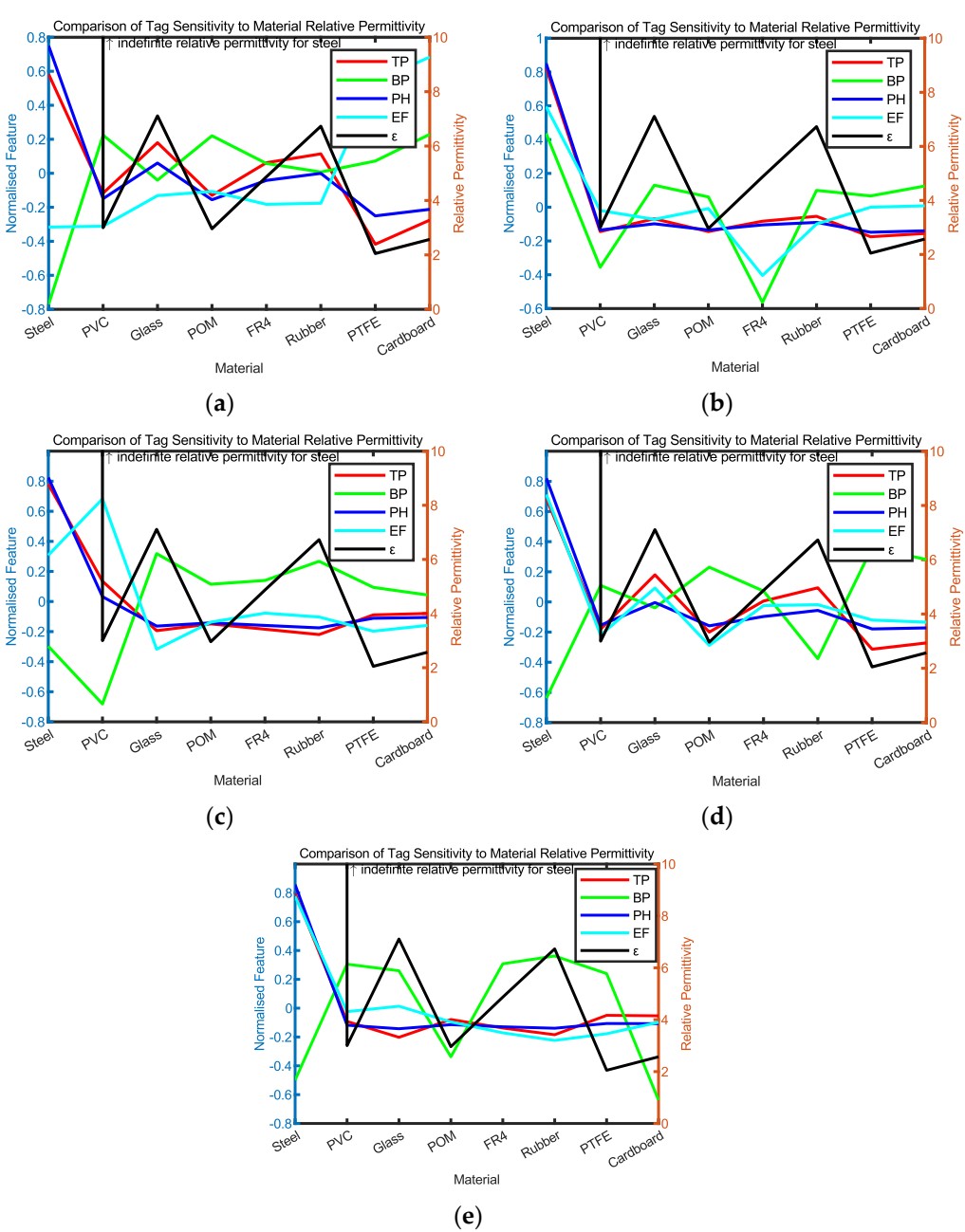

**Figure 7.** Measured parameters for different transponders (**a**–**e**).

The results for transponder a are shown in Figure 7. The first subplot shows the transmitted power, which is the minimum transmitted power from the reader to power on the transponder. This parameter is also known as the threshold power. This parameter can be used to identify the material as it shows diverse results for different materials, and no overlap occurred after 860 MHz. The difference between materials reached maximum at 880 MHz, then decreased as the operation frequency increased. The backscattered power was noisier compared with the transmitted power, and only the response signal for the steel stood out from other materials. Electric field strength showed a similar result as the transmitted power, but the difference between the steel sample and other samples was larger, indicating that this parameter was more sensitive to metal compared with the transmitted power parameter. The phase was unwrapped in the fourth subplot to change the angular change to the linear form, which complies with the form of other parameters. The phase feature shows that the PVC and POM had the highest phase variation across the band while those of glass, rubber, and FR4 were lower. The result of steel was among the lowest group, which means that the phase feature is less sensitive to metal detection.

For transponder b, there was a much smaller variance in the results for the non-metal objects. In contrast, the separation between metallic material and non-metallic material was the highest, which shows its potential for structural defect sensing.

The result of transponder c shows that this tag has a sensing band limited from 908 to 928 MHz, which is the same as its operation band. Its sensitivity to different materials is lower than transponder a's. Among all the parameters, transmitted power and electric field strength showed better sensitivity, while it was difficult to observe a constant trend in backscattered power.

Transponder d shows a lower sensitivity compared with transponder a, but it has a wider sensing band across the entire UHF band while using the transmitted power. The backscattered power was still noisy compared with the transmitted power or electric field strength, but the variance between different materials can be observed from the results.

The transmitted power parameter of transponder e shows a low sensitivity that was the fourth among the five tags. However, the phase feature of the transponder was similar to transponders a and c, which had the best sensitivity. The limitation of this tag was its size. With a length over 10 cm, transponder e will suffer from a non-flat surface in real applications.

The results show that the transmitted power is the best parameter for identifying the material using RFID and dipole-based RFID tags to achieve the best sensitivity. An unwrapped phase feature could provide better sensitivity for some antenna designs (transponder e), and it is more stable and immune to electromagnetic interference.

The results from raw features show that we can obtain different responses from different material samples. However, it is not easy to understand the relevance of these parameters with electromagnetic properties, such as the dielectric property and the dielectric loss. Thus, we use the transmitted power amplitude at the tag resonance in air to reveal the sensitivity of different tag designs to the relative permittivity, as shown in Figure 7. The dashed line in Figure 7 shows the relative permittivity of different materials, which is the same for Figure 8a. Other lines in Figure 8 show the reading from different tags. It is clear that tags b, c, and e show a similar trend as the change in relative permittivity between different material samples. This result can also be validated in the accuracy assessment of Figure 8b, which shows the correlation of the tag feature with the relative permittivity. Other parameters such as backscattered power and phase were also investigated for relative permittivity sensing, but no meaningful results could be achieved.

In the aspect of sensing the dielectric loss tangent, the backscattered power shows a superior sensitivity than other features, and this is shown in Figure 9. In this study, the backscattered power at the resonance of tag b shows the highest correlation with the change in dielectric loss tangent. However, tag a and tag d also show relevance to the change in dielectric loss, but negatively related.

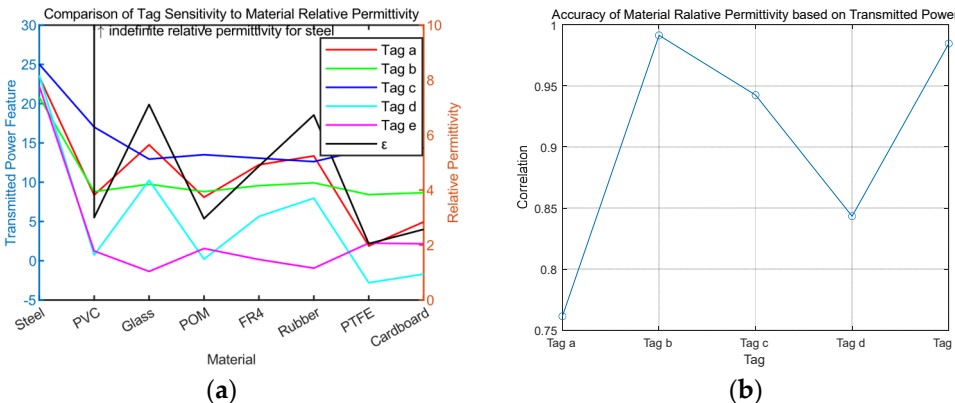

**Figure 8.** Transmitted power feature for relative permittivity measurement (**a**) results and (**b**) accuracy in correlation.

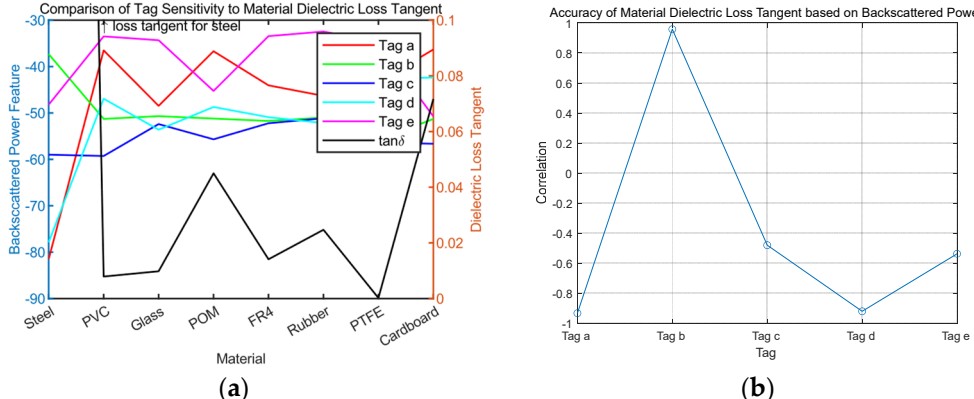

**Figure 9.** Backscattered power feature for dielectric loss tangent measurement (**a**) results and (**b**) accuracy in correlation.

To validate the effectiveness of machine learning techniques, we applied two traditional classification methods, tree and Naïve Bayes, and two MLNNs to extract the material information and tag used. We selected two-layer and three-layer MLNN in the study. The activation function used in the MLNN was ReLU (Rectified Linear Unit) for its advantages of better gradient propagation, and the iteration was limited at 1000 for the trade-off of efficiency. The neuron number was optimized to 66, which provides the highest accuracy. The accuracy of each method and processing time with parallel computing is shown in Table 3. The results show that Naïve Bayes has the worst accuracy for both identifying the material and the tag. MLNN shows the superior accuracy, with traditional methods with a relatively long processing time. Between the two MLNNs, the three-layer MLNN achieved a slightly even-better result. The tree algorithm also exhibited its application as it achieved acceptable accuracy within a short time, which inspires its application in distributed sensing conditions.

**Table 3.** Machine learning techniques accuracy for material identification.

| Method | Tree | Naïve Bayes | 2-Layer MLNN | 3-Layer MLNN |
|---|---|---|---|---|
| Accuracy | 64.4% | 23.4% | 77.9% | 78.9% |
| Time | 1.1821 s | 19.765 s | 102.32 s | 149.19 s |

The results show that the raw feature has an advantage in sensing single material property, achieving a high relevance with the property change. MLNN shows better classification accuracy compared with classic categorization methods, but the accuracy is not sufficient. This result helps the algorithm selection for sensing and classification applications.

### 3.4. Validation of Crack Sensing Capability of RFID

According to the results shown above, the tag used in this investigation was transponder b, which has the highest sensitivity and communication capability on metal and a low profile that does not cover two adjacent cracks during the experiment. The experiment setup was the same as introduced in the aforementioned section, except the sample was replaced with metal samples, which has artificial defects.

The results for the stainless steel sample are shown in Figure 10. Subplots (a) to (c) show the transmitted power, electric field strength, and unwrapped phase for the depth increase series. The resonant frequency of the tag antenna can be identified from the transmitted power, and it decreases in line with the crack depth increase. The amplitude of transmitted power can identify different depth cracks, but the relation is not linear. The electric field is similar to the transmitted power, providing the resonant frequency of the tag antenna. The absolute value of the electric field strength is also not linear with the growth of crack depth. The unwrapped phase results separate the different crack depth but also face the problem of a non-linear relationship. Thus, the resonant frequency feature is the best option for crack depth sensing using tag b. The results for the crack width series are shown in Figure 10b. The growth of the crack width also leads to the decrease of the resonant frequency shift of the tag antenna. An exception point is observed for the sixth crack for the width in TP, which is the sixth crack in series 2. The reason could be that the width of the crack introduces a new resonance at a higher frequency, and the combination of the two resonances increases the resonance. The length series results in Figure 10c show that the increase of the crack length also reduces the resonance of the tag antenna. As the resonance shows a decreasing trend for depth increase, width increase, as well as length increase, the next stage of work will be feature fusion with the non-linear features to distinguish these increases.

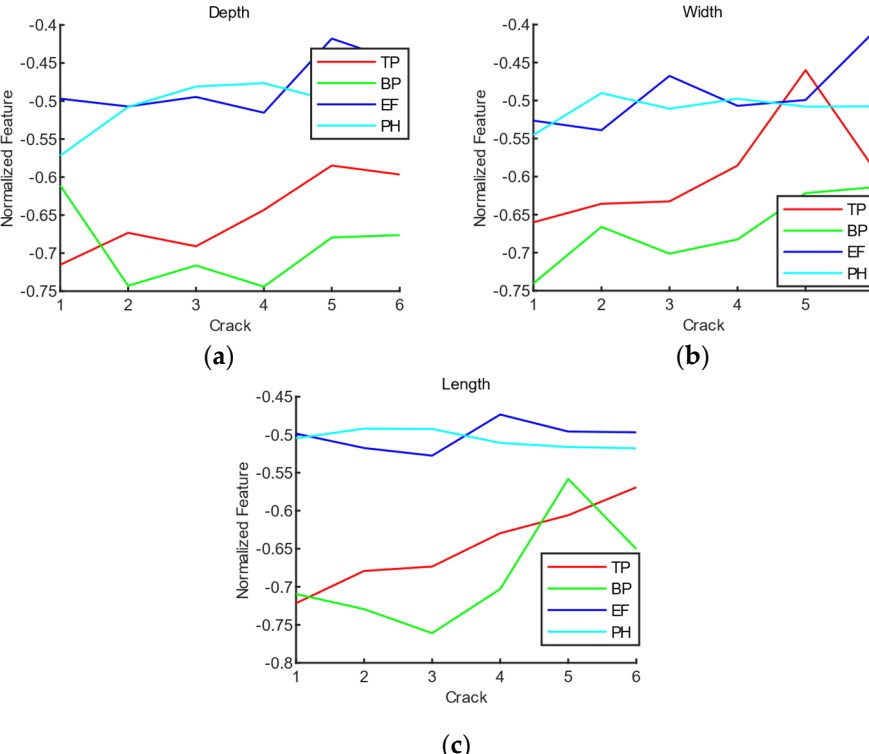

**Figure 10.** Stainless steel crack sensing results: (**a**) depth, (**b**) width, and (**c**) length.

The carbon steel results are present in Figure 11. Although both samples are metals, there are a few differences in the characteristics in the results. The sensor also shows a linear decrease for both increases in crack depth and length, but the sensor is more sensitive to depth increases in stainless steel and more sensitive to length increases in carbon steel.

Meanwhile, there is a risk of losing resonance on carbon steel when the width of the crack is between 3~3.5 mm. This results in losing the signal from the tag, affecting the reliability of the monitoring system.

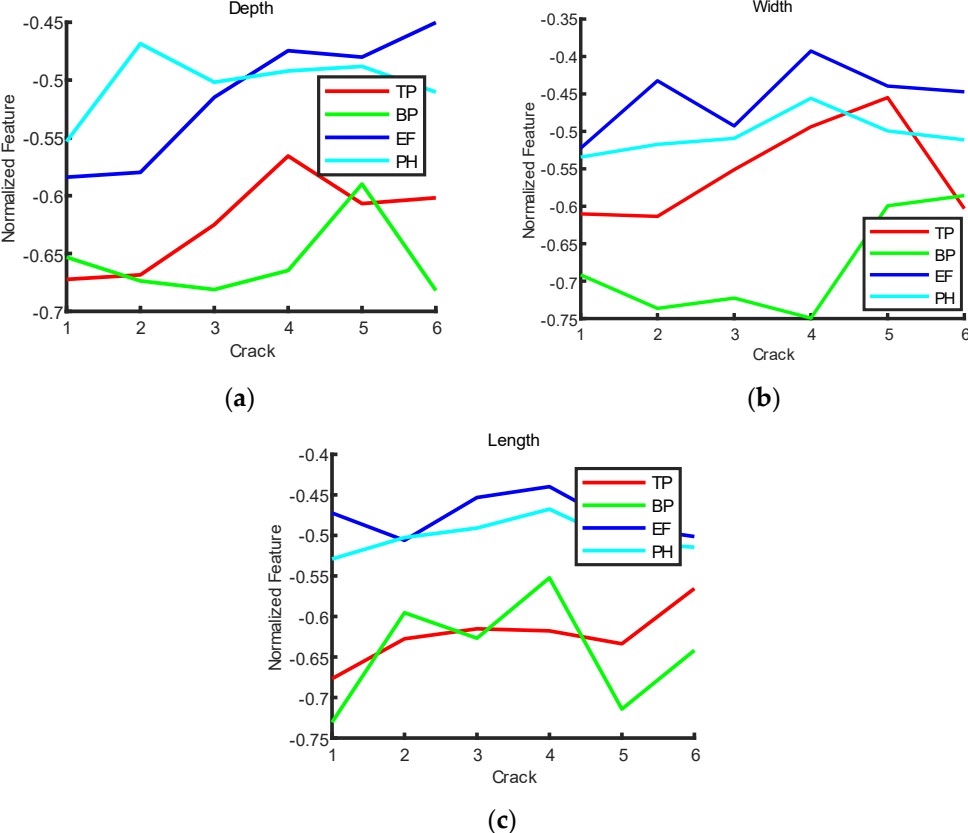

**Figure 11.** Carbon steel crack sensing results: (**a**) depth, (**b**) width, and (**c**) length.

Resonance frequency based on transmitted power is the most appropriate stand-alone feature, but the separation of the detailed crack characteristics is difficult to achieve with the raw features. Meanwhile, there are differences in the results from carbon steel and stainless steel, affecting the accuracy for sensing. Here, we applied the same feature extraction methods and MLNN to improve sensing accuracy in crack characterisation. The neuron number used was 65, the iteration limit as set as 1000, and the activation function was ReLU. The results are shown in Table 4. MLNN methods were still the most accurate methods. The tree method showed good performance in crack width sensing and extreme low processing time.

**Table 4.** Machine learning techniques accuracy for crack sensing.

| Crack Parameter | Tree | Naïve Bayes | 2-Layer MLNN | 3-Layer MLNN |
|---|---|---|---|---|
| Length | 82.1%, 1.4737 s | 66.4%, 14.027 s | 90.8%, 73.589 s | 91.3%, 111.05 s |
| Width | 88.2%, 1.2931 s | 65.2%, 8.3985 s | 94.2%, 70.958 s | 94.2%, 105.94 s |
| Depth | 79.8%, 1.5238 s | 48.0%, 14.052 s | 90.9%, 73.596 s | 91.4%, 107.57 s |

*3.5. Discussion*

To the best of the authors' knowledge, this was the first attempt to classify dielectric materials with RFID using ML techniques. One of the most widely used techniques for material dielectric property characterization is based on a novel microwave resonators, CSRR [48], which obtains an accurate measurement for permittivity (less than 0.1) and loss tangent (less than 0.0002). However, the setup is a direct contact measurement, which has limited applications in harsh environments, e.g., a nuclear decommissioning site.

This investigation also shows an improved accuracy for crack detection using MLNN methods with RFID compared with [29], which has an accuracy of 83.3% for a 0.5 mm width crack. Additionally, the full geometry sensing capability is presented, showing that the system has the best performance for crack width. This is because the sensing area of the RFID tag could cover the whole range of the crack width but only partially cover the length. Depth is a subsurface parameter and is, thus, less sensitive compared with the other two parameters. The results show that a future defect prediction and management system could consist of a centralised MLNN from the server side assisted with distributed tree algorithm-based sensing nodes for local and quick response.

## 4. Conclusions and Future Works

Non-destructive testing and evaluation (NDT&E) is vital for maintaining the safe operation of nuclear power plants or supporting POCO in decommissioning sites. In this paper, we have investigated an RFID-based sensing system for material classification and defect sensing integrated with machine learning techniques. This has not been investigated in the past. The results show that the raw feature is more effective in material property sensing compared with machine learning methods. Meanwhile, the machine learning method is effective in crack feature extraction, with a higher accuracy of 84.4% in width and relatively lower, at 78.7%, in depth. The theoretic model suggests that classic feature extraction methods have superior performance in measuring less variables, while MLNN has better performance when unknown factors are more numerous. Future works will focus on the selection guide for different applications regarding traditional methods and machine learning methods.

**Author Contributions:** Conceptualization, A.I.S. and A.Z.; Investigation, A.I.S. and A.Z.; Methodology, A.Z.; Validation, A.Z. and T.W.; Visualization, A.Z. and A.I.S.; Writing—original draft, A.Z. and A.I.S.; Writing—review and editing; A.I.S., T.W. and L.L.; Supervision, L.L. All authors have read and agreed to the published version of the manuscript.

**Funding:** The authors would like to acknowledge the financial support from the High Value Manufacturing Catapult (HVM Catapult) in the UK (Grant Number 160080).

**Institutional Review Board Statement:** Not applicable.

**Informed Consent Statement:** Not applicable.

**Data Availability Statement:** Data sharing not applicable.

**Conflicts of Interest:** The authors declare no conflict of interest.

## Abbreviation

| Acronyms | Definition |
| --- | --- |
| ANN | Artificial neural network |
| ASK | Amplitude shift keying |
| BP | Backscattered power |
| COTS | Commercial off-the-shelf |
| EF | Electric field |
| EMAT | Electromagnetic acoustic transducer |
| FR-4 | Flame retardant-4 |
| FS | Fuzzy logic-based system |
| HF | High frequency |
| IAEA | International Atomic Energy Agency |
| IC | Integrated circuit |
| IoT | Internet of Things |
| LF | Low frequency |
| MFL | Magnetic flux leakage |
| MHz | Megahertz |

| ML | Machine learning |
| MLNN | Multi-layer neural network |
| MRC | Magnetic resonance coupling |
| NDT | Non-destructive testing |
| NDT&E | Non-destructive testing and evaluation |
| PE | Processing element |
| PH | Phase |
| POM | Polyoxymethylene |
| POCO | Post Operation Clean Out |
| PTFE | Polytetrafluoroethylene |
| PVC | Polyvinylchloride |
| ReLU | Rectified Linear Unit |
| RFID | Radio frequency identification |
| SHM | Structural Health Monitoring |
| SVM | Support vector machines |
| TP | Transmitted power |
| UHF | Ultra-high frequency |

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
