# Peer review of "Machine Learning-Based Structural Health Monitoring Using RFID for Harsh Environmental Conditions"

_electronics, doi:10.3390/electronics11111740_

Round 1
Reviewer 1 Report
The title is interesting however, I didn't see any novelty of this work. Authors must be addressed clearly what authors newly presented in this research work also identify the compared existing tradition techniques. Comparison is required because readers are easily understood why this method is essential for harsh environmental conditions. Moreover, their is no clear information about the applications, how authors implemented the technique for harsh environmental conditions. Moreover, some of typo mistakes in the manuscript, example in conclusion authors were written " The theoretic model ". Abstract must be rewritten, no clear information in the abstract.
Author Response
The title is interesting however, I didn't see any novelty of this work. Authors must be addressed clearly what authors newly presented in this research work also identify the compared existing tradition techniques.
A: Thanks for pointing out this and we have added a brief introduction of the results in the abstract to clarify the novelty and contribution.
Comparison is required because readers are easily understood why this method is essential for harsh environmental conditions.
A: The reason of RFID suitability for harsh environment is modified as “The benefit of RFID is that it can operate in high temperature conditions [19] for its passive nature. Chip-based RFID can operate at 160 degrees [3] without the constraints from batteries while chip-less RFID is able to tolerant hundreds of degrees without an IC.” in the introduction.
Moreover, there is no clear information about the applications, how authors implemented the technique for harsh environmental conditions.
A: Thank you for the important query. There has numerous works done using RFID for NDT and SHM. However, one of the major challenges raises when there are multiple parameters to be inspected and in a harsh environment like nuclear power plant or decommissioning site. It is more evident that it is vital and very much desirable to have an automated system or application. Therefore, incorporating machine learning (ML) based health monitoring will vastly improve the system’s performance. This has been added in Section 1 Introduction and highlighted.
Moreover, some of typo mistakes in the manuscript, example in conclusion authors were written " The theoretic model ".
A: Thanks for pointing out the typo and the manuscript is reviewed and corrected.
Abstract must be rewritten, no clear information in the abstract.
A: A brief introduction of the results is added in the abstract to clarify the novelty and contribution.
Reviewer 2 Report
Post Operation Clean Out (POCO) is the process to remove hazardous materials and decommission nuclear facilities at the end of a nuclear plant lifetime. Introduction of Internet of Things (IoT) technologies in the environment would improve efficiency and safety by intelligently monitoring the POCO activities. The key technology for identification in IoT, Radio Frequency Identification (RFID), has been developed for sensing purpose and can achieve integration of sensing and communication.
The authors presented passive material identification and crack sensing method using commercial off the shelf (COTS) RFID tags, which is a long-term solution to material property monitoring under insulation for harsh environmental conditions.
The authors validated the effectiveness of material identification and crack monitoring, using machine learning techniques
The study has merits. However, there are some things to be revised and supplemented in writing the manuscript. I evaluate it as a manuscript that can be published in this journal if such things are supplemented. Those things are as follows:
- The abstract must be better harmonized to summarize each section in an equilibrated manner.
- Put a table with the acronyms
- Insert a clear purpose paragraph
- Equations must be revised and indexed
- The structure of the experimental validation must be revised. Introduce this section better. There is a first paragraph that is ok. Then there is a paragraph very short with a cryptic title “3.2. COTS RFID Tags”. Then there is a paragraph “Validation Results and Discussion”. The discussion is actually the description of the results.
- Insert a discussion with a comparison with other studies
- Check the resolution of the images and the figures.
Author Response
1. The abstract must be better harmonized to summarize each section in an equilibrated manner.
A: A brief description is added in the abstract to balance and summarize all sections.
2. Put a table with the acronyms
A: The acronyms table is added as Appendix A.
3. Insert a clear purpose paragraph
A: This has now been amended and added as “The focus of the paper is to explore the capabilities of ML in nuclear sector and this is done by carrying out a comparative study of identifying the correct material based on the ML approach and then determine varying cracks based on width, depth and length using both multilayer neural network (MLNN) and classic feature-based classification. The concept of this model involves a MLNN and its input features in terms of the backscattered power, transmitted power, electric field strength and unwrapped phase which are obtained from a RFID sensor. In order to identify the significance and future impact of the crack, the paper furthermore expands its objective by performing detection of both materials and cracks of different sizes. Such modelling essentially leads to a highly accurate system capable of identifying cracks and crack widths. Consequently, the UHF RFID sensor response can be proficiently interpreted to monitor infrastructure health and the decommission of the old reactors which is unavoidable at the end of their operation and POCO is carried out to tackle the radiation contaminated materials and ensure the safety and efficiency in the process.”
4. Equations must be revised and indexed
A: The equations are numbered accordingly.
5. The structure of the experimental validation must be revised. Introduce this section better. There is a first paragraph that is ok. Then there is a paragraph very short with a cryptic title “3.2. COTS RFID Tags”. Then there is a paragraph “Validation Results and Discussion”. The discussion is actually the description of the results.
A: 3.2 is renamed with “COTS RFID Tags for Validation” to clarify its content. 3.4 is added as a discussion for comparison with other studies.
6. Insert a discussion with a comparison with other studies
A: 3.4 is added as a discussion for comparison with other studies.
7. Check the resolution of the images and the figures.
A: This has now been checked and fixed.
Reviewer 3 Report
The authors have presented passive material identification and crack sensing methods using commercial off the shelf RFID tags. Although the proposed manuscript is well written, but the crack monitoring methods based on RFID sensing system employing machine learning is not new and the proposed work should be compared with previous methods and the advantages of the proposed one should be emphasized. The manuscript can be accepted after considering the following modifications.
- The equations, which are not extracted by the authors should be referenced in the manuscript. Also, all equations should be numbered in manuscript.
- Some abbreviations should be defined at the first seen in the text, such as SHM.
- In figure 2, three input parameters have shown, however five inputs are written, which are transmitted power, backscattered power, phase, frequency, and Electric field. Are all of these parameters considered for input parameters? If yes kindly show them correctly as the input of the model. Also, the other parameters of the proposed MLP network, such as the number of neurons, number of iterations, activation functions and etc. are not reported in the text.
- Kindly provide the equation, which is used for the accuracy results in Tables 3 and 4.
- Please explain why the machine learning technique accuracy for material identification is so low?
- Kindly provide a comparison between the obtained results and the other related methods for material identification or crack sensing
Author Response
1. The equations, which are not extracted by the authors should be referenced in the manuscript. Also, all equations should be numbered in manuscript.
A: The equations are numbered accordingly. And the referral model for deriving the equations is referenced as [44].
2. Some abbreviations should be defined at the first seen in the text, such as SHM.
A: All abbreviations are rechecked and a table is added as Appendix A for clarification.
3. In figure 2, three input parameters have shown, however five inputs are written, which are transmitted power, backscattered power, phase, frequency, and Electric field. Are all of these parameters considered for input parameters? If yes kindly show them correctly as the input of the model. Also, the other parameters of the proposed MLP network, such as the number of neurons, number of iterations, activation functions and etc. are not reported in the text.
A: The figure is revised and the model parameters are added as “We have selected two-layer and three-layer MLNN in the study and applied 10 neurons for each layer. The activation function used in the MLNN is ReLU for its advantages of better gradient propagation and the iteration is limited at 1000 for trade-off of efficiency.”.
4. Kindly provide the equation, which is used for the accuracy results in Tables 3 and 4.
A: The equation for the accuracy results of the material identification and crack sensing are provided individually in 2.5.
5. Please explain why the machine learning technique accuracy for material identification is so low?
A: Thanks for advice from the reviewer. We revisited the results with modified results with improved neuron and iteration numbers. The accuracy for material identification reaches 78.9% with neuron number of 66 for each layer with a three-layer MLNN at a maximum iteration number of 1000. However, this accuracy is still lower than the accuracy of crack sensing. The reason is the main influence of crack is the change of conductivity, which is a linear effect on the model. Different dielectric material induces change of both permittivity and loss tangent, result in more complicated model and lower accuracy. Further research will focus on refining the ML model with CNN and improve the material identification accuracy.
6. Kindly provide a comparison between the obtained results and the other related methods for material identification or crack sensing
A: The comparison is added as 3.4.
Round 2
Reviewer 1 Report
The authors addressed all of the comments except some grammatical typo mistakes in the manuscript. This revised can be accepted for publication.
Reviewer 2 Report
The manuscript impoved.
The authors extensively answered to the comments/suggestions.
The manuscript can be accepted.
There are not further comments.
Reviewer 3 Report
The authors have addressed all of my comments and the paper can be accepted in the present form.